# Multiphoton In Vivo Microscopy of Embryonic Thrombopoiesis Reveals the Generation of Platelets through Budding

**DOI:** 10.3390/cells12192411

**Published:** 2023-10-06

**Authors:** Huan Liu, Hellen Ishikawa-Ankerhold, Julia Winterhalter, Michael Lorenz, Mykhailo Vladymyrov, Steffen Massberg, Christian Schulz, Mathias Orban

**Affiliations:** 1Department of Internal Medicine I, Ludwig Maximilians University, 81377 Munich, Germany; huan.liu@med.uni-muenchen.de (H.L.); hellen.ishikawa-ankerhold@med.uni-muenchen.de (H.I.-A.); julia.winterhalter@med.uni-muenchen.de (J.W.); michael.lorenz@med.uni-muenchen.de (M.L.); steffen.massberg@med.uni-muenchen.de (S.M.); 2Laboratory for High Energy Physics (LHEP), Albert Einstein Center for Fundamental Physics, University of Bern, 3012 Bern, Switzerland; mykhailo.vladymyrov@unibe.ch; 3Theodor Kocher Institute, University of Bern, 3012 Bern, Switzerland; 4Data Science Lab, Mathematical Institute, University of Bern, 3012 Bern, Switzerland; 5DZHK (German Centre for Cardiovascular Research), Partner Site Munich Heart Alliance, 80802 Munich, Germany

**Keywords:** megakaryocyte, multiphoton intravital microscopy, thrombopoiesis, yolk sac, fetal liver, c-Myb

## Abstract

Platelets are generated by specialized cells called megakaryocytes (MKs). However, MK’s origin and platelet release mode have remained incompletely understood. Here, we established direct visualization of embryonic thrombopoiesis in vivo by combining multiphoton intravital microscopy (MP-IVM) with a fluorescence switch reporter mouse model under control of the platelet factor 4 promoter (*Pf4*^Cre^*Rosa26*^mTmG^). Using this microscopy tool, we discovered that fetal liver MKs provide higher thrombopoietic activity than yolk sac MKs. Mechanistically, fetal platelets were released from MKs either by membrane buds or the formation of proplatelets, with the former constituting the key process. In E14.5 c-Myb-deficient embryos that lack definitive hematopoiesis, MK and platelet numbers were similar to wild-type embryos, indicating the independence of embryonic thrombopoiesis from definitive hematopoiesis at this stage of development. In summary, our novel MP-IVM protocol allows the characterization of thrombopoiesis with high spatio-temporal resolution in the mouse embryo and has identified membrane budding as the main mechanism of fetal platelet production.

## 1. Introduction

Platelets play a crucial role in hemostasis in mammals [1]. In addition, they contribute to pathophysiological processes such as inflammation, infections, and cancer [2,3,4,5]. Platelets in adult mice are produced by megakaryocytes (MKs) in the bone marrow (BM). However, under thrombocytopenic conditions, platelet production is also allocated to the lungs [6]. The canonical MK generation model starts with hematopoietic stem-cell-derived megakaryocyte-erythrocyte progenitors (MEPs) differentiating into MK progenitor cells [7]. The latter eventually lose the capacity to divide and begin the process of endoreplication, leading to polyploidization and increases in cell size [8]. Mature MKs are located in the BM sinusoids [9], where they extend proplatelets into the blood circulation [10]. Proplatelets have periodic platelet-sized swellings, which are considered to mature into platelets within the circulation [11].

In contrast to platelet production in postnatal mammals, the mechanisms of embryonic thrombopoiesis have remained less well understood. The first platelets circulating in the fetus are provided by primitive megakaryocytes [12], which have a common progenitor with primitive erythrocytes in the yolk sac (YS) [13]. These MEPs differentiate into MK progenitor cells [7]. The first MK wave is detected in the YS and is characterized by the occurrence of diploid MKs [13]. MKs with higher ploidy are uncommon at this developmental age. The second wave of hematopoiesis in the YS and subsequently fetal liver (FL) gives rise to MKs that are able to polyploidize but are still smaller than adult MKs. The proportion of newly formed, RNA-rich reticulated platelets is higher in the fetus compared to the adult organism [14,15]. Although these various details of fetal thrombopoiesis have been identified, the mode of embryonic platelet production from MKs has remained unclear. There are reports that, besides conventional proplatelet generation from long membrane protrusions, another form termed proplatelet budding from MKs can be observed in murine embryos [16]. Understanding and quantifying the mechanism by which platelets are generated at the embryonic stage could be critical to developing novel therapeutic strategies for congenital defects of hematopoiesis and thrombopoiesis, like neonatal thrombocytopenia.

As such, the dynamics of MK’s development and their mode of embryonic platelet production have not been studied in detail by real-time in vivo imaging, which offers detailed insights into biologic processes [17]. Therefore, we developed a multiphoton intravital microscopy (MP-IVM) protocol suitable to visualize platelet biogenesis from their MK progenitors in real time with high spatio-temporal resolution. It allowed us to perform extensive quantitative analysis of embryonic platelets and MKs in blood, YS, and FL over different stages of murine development to give a detailed view on embryonic thrombopoiesis.

Blood formation relies on transcription factors driving hematopoietic differentiation. The transcription factor c-Myb promotes survival, proliferation, and differentiation of hematopoietic cells, and it is therefore essential for the establishment of definitive hematopoiesis [18,19]. Early hematopoiesis in the mouse embryo is, however, independent of c-Myb, thus allowing for the distinction of early and later (definitive) stages of hematopoiesis [20]. The consequences of Myb deficiency become apparent from embryonic day E14 onwards, at which the mouse embryos develop anemia due to the lack of mature erythrocytes. Consequently, Myb-knockout embryos die with anemia around E16.5 [20]. We have shown previously that YS hematopoiesis provides blood cell lineages independently of c-Myb [21]. In line with this, YS megakaryocytes can derive from erythro-myeloid progenitors at this stage [22]. Here, we utilized c-Myb transgenic mice to determine the association of platelet production with early hematopoiesis in the mouse embryo.

In summary, we combined time-lapse MP-IVM with three-dimensional volumetric imaging (4D imaging) to visualize embryonic thrombopoiesis in hematopoietic organs (YS, FL) across different stages of embryonic development. By these means, we identified platelet budding as the major process in embryonic thrombopoiesis.

## 2. Materials and Methods

### 2.1. Mice

C57BL/6, *Pf4*^Cre/wt^ (Stock No: 008535) [23], and *Rosa26*^mTmG/wt^ mice (Stock No: 007676) [22] were obtained from The Jackson Laboratory (Boston, MA, USA). Myb^−/−^ mice (Stock No: 004757) have previously been described [19]. The mice were maintained and bred in specific, pathogen-free environments and fed a standard mouse diet. All pregnant animals undergoing procedures were over 10 weeks old.

Embryonic development was estimated as embryonic day 0.5 (E0.5) if a vaginal plug was present the following morning after mating. Pregnant females were used for subsequent experiments at indicated time points. Abdominal ultrasounds were carried out to monitor pregnancy. Embryos of E11.5 to E14.5 were adopted for flow cytometry analysis. Male *Rosa26*^mTmG/wt^ × *Pf4*^Cre/wt^ mice were mated with female C57BL/6 mice to generate heterozygous *Rosa26*^mTmG/wt^ × *Pf4*^Cre/wt^ embryos expressing the fluorescent marker mTmG. As controlled by activity of the Pf4 promoter, Cre activity leads to fluorescence switch from constitutive tomato (mT) to green fluorescent protein (mGFP) [23,24]. MP-IVM was performed for YS of E9.5–E10.5 and FL of E13.5–E15.5 (Appendix A). Mice were sacrificed after surgery. All procedures were approved by the local legislation on protection of animals. Animal studies were approved by the local regulatory agency (55.2-1-54-2532-207-2016, 55.2-1-54-2532-181-13).

### 2.2. Multiphoton Intravital Microscopy for Yolk Sac and Fetal Liver Imaging

#### 2.2.1. Preparation of Animals for Yolk Sac and Fetal Liver Imaging

Surgery of pregnant mice was performed under deep anesthesia by intraperitoneal injection of midazolam (5 mg/kg body weight; Panpharma GmbH, Trittau, Germany), medetomidine (0.5 mg/kg body weight; Laboratorios Syva S.A.U., León, Spain), and fentanyl (0.05 mg/kg body weight; Eurovet Animal Health BV, AE Bladel, Netherlands). During the subsequent operation, surgical tolerance of anaesthetized mice was checked regularly. Mice were placed in supine position on surgical heating stage. The abdominal skin was disinfected with antiseptic solution (Octenisept^®^, Norderstedt, Germany). A 1.5 cm skin incision was made on the lateral side of the abdomen, and subsequently, the underside portion of the peritoneal membrane was cut. A single uterine horn containing embryos was carefully exposed for dissection of the uterus to uncover the embryo. The uterine wall was cut with the spring scissor, revealing the individual amniotic sacs, each containing one fetus and the respective placenta. For YS imaging, this specific uterine area was placed and fixed on a holder that was custom-made by 3D printing. The membranes surrounding the YS were carefully removed and cleaned. Warm ultrasound gel was added around the embryo and YS to assure tissue hydration. A holder with coverslip was placed on top of the exposed YS. Then, the mouse preparation was moved to the microscope stage. For FL imaging, the YS and amniotic sac were cut to expose the fetus. The fetus was placed and fixed on the lower holder. The lateral abdominal skin of the embryo was removed to reduce light scattering and improve light penetration. Identically, the holder with coverslip was placed on the top of the exposed FL. During the whole preparation and following imaging process, warm saline or ultrasound gel (SONOSID^®^, Herrenberg, Germany) was continuously added to keep the hydration of the exposed uterus and embryo. The embryo was maintained in connection with the mother through the umbilical cord to preserve its circulation.

The coverslip holders were redesigned by 3D printing technology so that the holders and coverslip size could be adjusted according to the embryonic age and size. We have designed the different holder’s sizes using computer-aided design (CAD) software (Autodesk^®^ Fusion 360™, Version 16.4.0.2062), a biocompatible plastic resin (MiiCraft BV-007A), and a 3D printer (MiiCraft-Ultra 50, Burms-Germany).

#### 2.2.2. Thrombopoiesis Multiphoton Intravital Imaging in the Yolk Sac and Fetal Liver

The microscope stage carrying the mother animal and the exposed and connected embryo were placed under the microscope objective. GFP-expressing cells were identified by epifluorescence mode (X-Cite 120Q) before image scanning was carried out using the MP-IVM mode. Only Pf4 Cre-transgenic embryos expressed the GFP reporter in the megakaryocyte/platelet lineage. Cre-negative embryos were further used to confirm specificity of the fluorescence signal.

A multiphoton microscope (Trim-Scope II- LaVision Biotech, Bielefeld, Germany) with an upright Olympus stand, all enclosed in a custom-built incubator maintaining 37 °C, was used for imaging. The microscope is equipped with a Chameleon Ti:Sapphire laser (Coherent, Inc., Santa Clara, CA, USA) with an output 900–1100 nm range pulsed laser for excitation and four photomultipliers (PMTs) with four simultaneous channel acquisitions. Images were acquired using a CFI75 LWD 16X objective (Nikon) with saline (for YS) or ultrasound gel (for FL) as immersion medium. Emitted fluorescent signals were collected by photomultipliers. Images were acquired using ImSpector software (Version ImspectorPro 275). Fluorescence excitation was provided by laser tuned to 1100 nm with an optical parametric oscillator (OPO) for simultaneous excitation of Tomato fluorophore. The 800 nm excitation wavelength was used for imaging GFP. The power of the laser and OPO were adjusted according to the thickness of the scan. The magnifications were described for each specific image. Image stacks (*z*-axis) were recorded with 1.5 μm or 2 µm step size with the axial depth of 20–90 μm tissue volume. Sequential images were acquired as time lapse to allow observation of fetal blood flow.

#### 2.2.3. Video Stabilization and Analysis

During MP-IVM imaging acquisition, VivoFollow (Version 02) life drift correction software tool was employed to correct the tissue motion in real-time and maintain the region of interest in focus over the course of imaging [25,26]. After MP-IVM, all the acquired in vivo real-time imaging videos were analyzed with the dedicated software Imaris (Version 8.4.2) (Bitplane), enabling advanced video processing and in-depth analysis of particles. Vessels were defined as low-contrast (dark) regions with intravascularly flowing cells. Membrane budding as a form of platelet generation was defined if the extension or membrane protrusion was less than 10 µm in length before its release from the MK. Proplatelet formation was referred to as a membrane protrusion longer than 10 µm. The fraction of each type of platelet generation mode was calculated for all visualized MKs. Optional post-acquisition image processing enabled further stabilization of tissue motion by additional drift correction.

### 2.3. Flow Cytometry

#### 2.3.1. Blood Collection, Processing, and Fluorescence Staining

For platelet and reticulated platelet analysis, maternal and embryonic blood were taken (Appendix A). Full blood from the mother mice was taken by cardiac puncture with a syringe containing acid-citrate-dextrose (ACD, containing 39 mM citric acid, 75 mM sodium citrate, and 135 mM dextrose; 1:7 volume). The blood sample was kept at room temperature for further procedures. Embryos were isolated from pregnant mice at indicated timepoints. Collection of 2 µL of fetal blood was performed through lateral neck dissection and added to 98 µL PBS for platelet and reticulated platelet staining. Platelets within a total volume of 100 µL were stained with the fluorescent-labeled antibody anti-CD42d-APC (Biolegend, San Diego, CA, USA) for 25 min at room temperature. The sample was then diluted again with PBS to 400 µL. For platelet counting, 4 × 10^4^ beads (Thermo Fisher, Waltham, MA, USA) were added before flow cytometry. Platelet counts were measured from 2 µL blood volume by flow cytometry. Peripheral platelets from embryonic and maternal blood as a control were gated with the same gating strategy (Appendix A). Platelets were defined as (1) low signal in forward scatter/sideward scatter (FSC/SSC), (2) singlets by FSC width, and (3) CD42d positive [6]. Platelet counts were calculated based on collected volume and counted number of cells. For reticulated platelet staining, 2 µL of blood was fixed with PFA 1%. Then, samples were stained with anti-CD42d-APC and thiazole orange (TO) (1 μg/mL) [27] (SigmaAldrich, Poole, UK) for a further 25 min at room temperature. Embryonic and maternal blood samples were submitted to flow cytometry analyses and gated with additional gating for TO-positive particles representing reticulated platelets (Appendix A).

#### 2.3.2. Fetal Liver Isolation, Processing, and Fluorescence Staining

The fetal liver was isolated from surrounding tissue by blunt dissection and set aside in a solution of saline (NaCl 0.9%) on ice (Appendix A). With a 5 mL syringe paired with a 20-gauge needle, the fetal liver was aspirated and expelled several times to mechanically dissociate the tissue. The dissociated liver was filtered through a 70 µm cell strainer and resuspended for subsequent cell counting and flow cytometry staining. Fetal liver cell numbers were counted with a Neubauer counting chamber. Cell staining and flow cytometry analysis were carried out as previously described [28]. For MK fraction assessment, the aforementioned cell suspension was stained with fluorescent-labeled antibodies anti-CD41-FITC (Thermo Fisher Scientific Company, San Diego, CA, USA) and anti-CD42d-APC. After washing and resuspension, cells were transferred for flow cytometry. MKs from FL were defined as (1) large FSC/SSC cells, singlets in FSC-H/FSC-W and double-positive for the thrombopoietic surface markers CD41 and CD42d [29] (Appendix A). Propidium iodide (PI) (Sigma-Aldrich, Taufkirchen, Germany) was used to analyse ploidy. After staining with anti-CD41-FITC and anti-CD42d-APC, the cell resuspension was fixed with PFA 1%. After fixation, cells were washed and resuspended with 500 µL DNA staining buffer (1×PBS + 2mM MgCl_2_ + 0.05% Saponin + 0.01mg/mL PI + 10U/mL RNase A) overnight at 4 °C. The aforementioned MK gating strategy in flow cytometry was used to identify MKs. Histograms were divided to approximate the different ploidy classes in terms of DNA content. We set boundaries of DNA with markers between peaks with geometric mean analysis [30].

#### 2.3.3. Flow Cytometry Acquisition and Analysis

Samples were analyzed on the flow cytometer (Beckman Coulter, Gallios Flow Cytometer 773231AD, Diva software version 8.0.1). Acquired flow cytometry data sets were analyzed with a dedicated flow cytometry analysis software (FlowJo LLC Version 10, Ashland, OR, USA).

### 2.4. Statistical Analysis

The fraction of MKs was calculated as the number of the respective MK type (e.g., intravascular) divided by the total number of MKs observed in the volume. The concentration of bud-releasing MKs was calculated as follows: MK number was divided by observation time and volume scanned, which was expressed as MK/(h×mm^3^). Visualization and statistical analyses were performed with GraphPad Prism software Version 9 (San Diego, CA, USA). In figures, all data are presented as mean ± SD. An unpaired student′s *t*-test was used for comparison between two groups. For comparison of multiple groups with one variable, one-way ANOVA testing was performed. A *p* value less than 0.05 was considered statistically significant, which was indicated by ‘*’ in figures (* *p* < 0.05).

## 3. Results

### 3.1. The 4D Imaging of MKs in YS and FL Sinusoids

To study embryonic thrombopoiesis in vivo, we established a protocol for the intravital visualization of platelet generation from MKs over time with MP-IVM in *Pf4*^Cre/wt^ and *Rosa26*^mTmG/wt^ embryos. We imaged platelet generation from MKs in YS and FL, identifying various types of proplatelets at different stages of embryonic development.

Using MP-IVM, we visualized hematopoietic cells as well as parenchymal and vascular structures within YS and FL. MKs expressing GFP presented with different sizes and morphologies and were distributed intravascularly and extravascularly (Figure 1A). To evaluate the density of MKs in YS and FL sinusoids, we measured the number of MKs per mm^3^ of the imaged 3D dataset from MP-IVM (Figure 1B). In YS, there were more MKs at E10.5 (1045 ± 602 cells per mm^3^) than E11.5 (499 ± 326 cells per mm^3^, *p* = 0.009). Still, the MK density was higher in FL than in YS. In FL, MK density at E13.5 was 1621 ± 798 cells per mm^3^ and at E14.5 1519 ± 655 cells per mm^3^. At E15.5, FL MK density decreased to 535 ± 303 cells per mm^3^. In YS, the diameter of MKs was 13.6 ± 1.6 µm at E10.5, which was similar to 14.8 ± 3.1 µm at E11.5. In FL at E13.5, MK diameter was 21.4 ± 0.9 µm. This was comparable to E14.5 with 21.5 ± 2.6 µm. At E15.5, the MK diameter increased to 24.4 ± 1.9 µm (Figure 1C). Accordingly, FL MK was bigger than YS MK.

We further evaluated the spatial association of MKs with the vasculature within both hematopoietic compartments of YS and FL. We used MP-IVM to determine the positioning of maturing MKs in the fetal vascular compartment. From MP-IVM 3D datasets, we quantified MK positions in the vasculature of YS and FL and categorized into intravascular, crossing the vessel wall, or extravascular MK (Figure 1D). In YS, the fraction of intravascular MKs at E10.5 and E11.5 were similar (89.2 ± 16.6% vs. 81.6 ± 19.3%, *p* = ns). In FL, the fraction of intravascular MKs was very low, from E13.5 to E14.5 (5.5 ± 6.9% and 3.6 ± 4.5%). At E15.5, the fraction of intravascular MKs was higher (14.6 ± 17.1%) (Figure 1E left). As for vessel wall-crossing MKs, their fraction was low in YS at E10.5 (6.3 ± 11.7%) and at E11.5 (1.5 ± 3.3%). This fraction was significantly higher in FL at all observed days, with a constant increase over time (E13.5 18.5 ± 5.2%, E14.5 19.5 ± 12.7%, and 41.3 ± 30.0%, Figure 1E middle). Extravascular MKs were a minority in YS at E10.5 (5.0 ± 13.6%) and at E11.5 (14.0 ± 21.7%). As such, the fraction of extravascular MKs in YS was lower than in FL. Extravascular FL MKs were the majority, but with declining fractions from E13.5 to E14.5 and E15.5 (76.1 ± 7.5%, 76.9 ± 13.9%, and 44.2 ± 27.7%) (Figure 1E right).

### 3.2. Different Forms of Platelet Shedding from Fetal MKs

We observed two distinct forms of potential platelet shedding from MKs by MP-IVM: (i) membrane budding and (ii) proplatelet formation. Detailed observation of MKs revealed a different form of platelet generation by expelling smaller particles without long proplatelet formation, which we termed the MK releasing bud (Figure 2A and Appendix A). As reported in adult bone marrow [10], embryonic MKs also showed conventional proplatelet formation by extending long membrane protrusions into the vasculature (Figure 2B and Appendix A).

“MK forming” bud means that we quantified all MKs’ assembling cytoplasmatic protrusions of less than 10 µm. In addition, we quantified those MKs that released buds, using the term “MK releasing buds”. We observed a significant number of MKs forming protrusions, including releasing events through buds and proplatelets (Figure 2C). In YS, there were 310.7 ± 486.7 bud-forming MKs/h×mm^3^ at E10.5 and 238.9 ± 344.1 bud-forming MKs/h×mm^3^ at E11.5. Correspondingly, there were 118.8 ± 228.8 proplatelet-forming MKs/h×mm^3^ at E10.5 and 19.3 ± 61.1 proplatelet-forming MKs/h×mm^3^ at E11.5. In FL, there were 506.7 ± 228.8 bud-forming MKs/h×mm^3^ at E13.5 and 358.9 ± 206.8 bud-forming MKs/h×mm^3^ at E14.5. Correspondingly, there were 52.0 ± 85.6 proplatelet-forming MKs/h×mm^3^ at E13.5 and 76.1 ± 114.5 proplatelet-forming MKs/h×mm^3^ at E14.5. The same trend appeared at E15.5, as we quantified more bud-forming MKs (271.7 ± 118.8 MKs/h×mm^3^) than proplatelet-forming MKs (59.6 ± 87.8 MKs/h×mm^3^). In YS at E10.5, the fraction of bud-forming MKs was numerically higher than proplatelet-forming MKs (11.4 ± 12.8% vs. 5.8 ± 12.6%, *p* = 0.26) (Figure 2D). However, at E11.5, the fraction of bud-forming MKs (18.5 ± 20.6%) was much higher than the fraction of proplatelet-forming MKs (0.89 ± 2.5%, *p* = 0.03). In FL, the fraction of bud-forming MKs was higher than the fraction of proplatelet-forming MKs at E13.5 (32.2 ± 11.2% vs. 3.5 ± 2.6%, *p* = 0.0009), at E14.5 (29.1 ± 5.4% vs. 3.1 ± 4.7%, *p* = 0.0009), and at E15.5 (42.0 ± 14.7% vs. 6.7 ± 9.2%, *p* < 0.0001).

Next, we quantified how many MKs were releasing buds or proplatelets per hour per mm^3^-scanned volume (Figure 2E). In YS, there were 77.0 ± 193.5 bud-releasing MK/h×mm^3^ at E10.5 and 83.0 ± 194.2 bud-releasing MK/h×mm^3^ at E11.5. Correspondingly, there were 20.0 ± 84.8 proplatelet-releasing MK/h×mm^3^ at E10.5 and 21.5 ± 64.4 proplatelet-releasing MK/h×mm^3^ at E11.5. In FL, there were 239.0 ± 186.5 bud-releasing MK/h×mm^3^ at E13.5 and 132.3 ± 87.1 bud-releasing MK/h×mm^3^ at E14.5. There were 12.6 ± 28.2 proplatelet-releasing MK/h×mm^3^ at E13.5 and 9.7 ± 25.6 proplatelet-releasing MK/h×mm^3^ at E14.5. At E15.5, there were 151.8 ± 74.7 bud-releasing MK/h×mm^3^, and no proplatelet-releasing was observed yet. We quantified the percentage of bud-releasing and proplatelet-releasing MKs among the MK population (Figure 2F). In YS, the fraction of MKs with membrane bud release (3.0 ± 7.2%) was similar to proplatelet release (1.2 ± 4.9%) at E10.5. At E11.5, there was a numerical increase in membrane bud release (4.8 ± 8.9%) compared to E10.5. The difference between membrane bud release and proplatelet release increased from E10.5 to E11.5. The proplatelet release fraction (0.9 ± 2.5%) at E11.5 was similar to E10.5. In FL at E13.5, the fraction of MKs with membrane bud release (12.9 ± 11.5%) was 10-fold higher than proplatelet release (0.9 ± 2.3%, *p* = 0.03). This difference was also present at E14.5 (11.9 ± 10.4% vs. 0.4 ± 1.1%, *p* = 0.01) and E15.5 (25.4 ± 14.4% vs. 0.0 ± 0.0%, *p* < 0.0001). In both YS and FL, platelet release events were rare. This is in line with a former report [16] pointing at a larger abundance of bud release. Overall, there were more bud-release events than proplatelet-release events both in YS and FL. The fraction of bud-forming MKs was higher in the FL than the YS. This indicated higher thrombopoietic activity in the developing FL over time.

### 3.3. Thrombopoietic Activity Profile

Whether the aforementioned increase in bud-release events from MKs translated into an increase in peripheral platelets was subsequently investigated. As is known, the major source of MKs in embryonic development is the fetal liver [31,32]. We measured the platelet count in embryo peripheral blood from E11.5 to E14.5 by flow cytometry (Figure 3A). At E11.5, the blood platelet count was the lowest (8.1 ± 4.6 × 10^3^/µL), with a sharp increase at E12.5 (83.2 ± 45.4 × 10^3^/µL) and continued growth to E14.5 (165.3 ± 78.1 × 10^3^/µL) (Figure 3B). The peripheral blood platelet count in maternal blood remained stable at different stages of gestation (Appendix A).

Newly formed platelets, so-called reticulated platelets, are anucleated, but they contain rough endoplasmic reticulum and messenger RNA. Reticulated platelets exist in adult mouse blood and can rise in numbers under certain conditions [33]. With flow cytometry, we qualified the reticulated platelet percentage (Figure 3C). The majority of embryonic platelets were reticulated from E11.5 until E14.5 (Figure 3D). At E11.5, the fraction amounted to 79.4 ± 13.4% of all blood platelets. With embryonic development, the proportion decreased continuously to 51.4 ± 7.7% at E14.5 (*p* < 0.0001). For the mother mice, the reticulated platelet fraction ranged from a minimum of 8.2 ± 2.9% at G13.5 to a maximum of 12.5 ± 2.2% at G11.5, although these differences over gestational age did not reach statistical significance (*p* = 0.34, one-way ANOVA test) (Appendix A). These results indicate a higher level of thrombopoietic activity in the embryo.

### 3.4. MK Development within the Fetal Liver Niche

Between E11.5 and E15.5, the fetal liver undergoes a period of accelerated growth with rising FL cell numbers and density [34]. FLs from embryos of different ages (*n* = 10–21 embryos and *n* = 3–5 mothers) were analyzed, showing an increased total cell number from 0.04 ± 0.05 × 10^6^ cells at E11.5 to 19.7 ± 7.4 × 10^6^ cells at E14.5 per FL (Figure 4A).

Using flow cytometry, we quantified MKs in FL (Figure 4B) in different embryonic stages. The absolute MK numbers per FL were calculated from the FL cell count multiplied by the fraction of MK in FL. During FL development, total cell numbers strongly increased. In parallel, fetal MK numbers increased significantly (Figure 4C). The absolute MK counts grew from 0.2 ± 0.3 × 10^3^ at E11.5 to 10.8 ± 4.2 × 10^3^ at E14.5 (*n* = 10–21 embryos and *n*= 3–5 mothers).

The maturation of fetal liver MKs was assessed with ploidy analysis using propidium iodide (PI) staining measured by flow cytometry across embryonic development (Figure 4D). The 2N MK fraction was highest at E11.5 with 43.3 ± 4.9% and decreased until E13.5 (20.7 ± 1.8%). The 4N MK fraction was the dominant fraction at different embryonic ages and remained stable. At E11.5, the 4N MK fraction was 48.3 ± 21.0%. The 8N MK fraction was low at E11.5 (0.3 ± 1.1%) and increased to 6.8 ± 4.8% at E12.5 and 22.4 ± 9.7% at E13.5. E14.5 is similar to E13.5 (21.5 ± 7.0%). The 16N MK fraction increased from E11.5 (0.4 ± 1.2%) to E13.5 (3.4 ± 2.2%) and E14.5 (2.9 ± 1.9%). The 32N and 64N MK fractions were very small (less than 1%) throughout embryonic development (Figure 4E). In summary, the increase in the 8N MK fraction paralleled with a decrease in the 2N fraction could hint at MK maturation over time in the FL.

### 3.5. Impact of c-Myb on Embryonic Thrombopoiesis

Next, we evaluated the role of primitive and definitive hematopoiesis regulated by c-Myb in embryonic thrombopoiesis. MKs in embryos are believed to originate from both primitive hematopoiesis and definitive hematopoiesis [35]. Homozygous c-Myb^−/−^ embryos lack definitive hematopoiesis and die around day E16.5 with severe anemia [20]. We used a c-Myb transgenic mouse model and measured platelet and MK levels from littermates at E14.5 by flow cytometry to evaluate the effect of c-Myb and therefore definitive hematopoiesis.

We measured platelet counts of different c-Myb genotypes at E14.5: Homozygous c-Myb^−/−^ embryos were compared to heterozygous c-Myb^+/−^, and c-Myb^+/+^ embryos. Embryonic platelet count in c-Myb^−/−^ embryos (190.0 ± 25.6 × 10^3^/µL) was similar to c-Myb^+/+^ (173.3 ± 37.9 × 10^3^/µL), but both numbers were lower than heterozygous c-Myb^+/−^ (287.3 ± 16.1 × 10^3^/μL) (Figure 5A).

Subsequently, we evaluated the fraction of reticulated platelets in embryos with different c-Myb genotypes. Reticulated platelet fraction was comparable in c-Myb^−/−^ (56.5 ± 3.4%) and c-Myb^+/+^ (52.2 ± 3.8%) mice (Figure 5B). These results show that c-Myb does not seem to have a profound effect on reticulated platelet generation at E14.5.

Since the size of the FL in c-Myb^−/−^ embryos was smaller due to the lack of definitive hematopoiesis, FL cell counts were significantly lower in c-Myb^−/−^ (1.8 ± 1.4 × 10^6^ per fetal liver) than c-Myb^+/+^ (16.3 ± 6.2 × 10^6^ per fetal liver, *p* = 0.007) embryos (Figure 5C). When investigating MKs, c-Myb^−/−^ embryos displayed absolute numbers comparable to c-Myb^+/+^ embryos (12.8 ± 8.5 × 10^3^ MKs per fetal liver vs. 9.8 ± 3.4 × 10^3^ MKs per fetal liver) (Figure 5D). Ploidy analysis revealed that the majority of MKs were 2N and 4N in all embryos (Figure 5E). The 2N fraction was lowest in c-Myb^+/+^ MKs (17.7 ± 1.5%) and highest in c-Myb^−/−^ MKs (35.9 ± 31.0%). The 4N fraction was similar across all genotypes (between 44.8 ± 12.1% and 45.5 ± 1.3%). In contrast to 2N, the 8N fraction was lowest in c-Myb^−/−^ MKs (16.8 ± 16.6%) and highest in c-Myb^+/+^ (31.8 ± 1.0%). The 16N, 32N and 64N fractions were very low in all genotypes.

Even though c-Myb embryos had lower FL cell numbers, they presented a similar level of platelets and their reticulated forms in peripheral blood. This could be explained by their similar absolute FL MK numbers compared with c-Myb^+/+^ embryos. These results confirmed that c-Myb had a strong impact on definitive haematopoiesis as FL cell numbers were very low. Additionally, c-Myb had no apparent impact on thrombopoiesis until E14.5. These findings could hint to at the potential of primitive thrombopoietic cells for maintaining platelet generation in the absence of definitive hematopoiesis.

## 4. Discussion

Using MP-IVM, we determined active thrombopoiesis in YS and FL. Comparing the different sites of thrombopoiesis in live embryos, we found higher density, increased ploidy, and a larger size of MKs in FL than in YS. This indicates higher differentiation and thrombopoietic activity in FL, reflecting a higher demand for circulating blood cells in the fast-growing embryo. Nevertheless, MK density decreases over time in both organs, likely due to the expansion of other hematopoietic lineages such as red blood cells. Regarding spatial distribution, the majority of MKs in YS were located in the intraluminal area of the vessels. On the contrary, in FL, a large fraction of MKs were positioned in the extravascular space. Whether this is due to a role of MKs in niche homeostasis as described for adult bone marrow, requires further investigations in future studies [36,37,38].

Using MP-IVM, we revealed the diverse MK and proplatelet/platelet populations present during embryonic development. Compared with the classic proplatelet formation and shedding pattern in adult bone marrow and lung [6,10], we found small cytoplasmatic extensions, which we termed buds. The latter were released and directly shed into circulation from FL and YS MKs and represented the majority of release events. It is noteworthy that in vitro, the vast majority of MKs do not form buds but long proplatelets. The discrepancy between these in vitro and in vivo results has been shown by others recently [16]. Furthermore, when canonical proplatelet protrusions were observed, they were shorter than their adult counterparts [39]. Importantly, the modes of platelet generation seem to be different between prenatal and adult life. Brown et al. found that most megakaryocytes enter the sinusoidal space as large protrusions rather than extruding fine proplatelet extensions [40].

The potential mechanisms of budding vs. proplatelet shedding remain incompletely understood. Surely, the cytoskeleton and membrane re-organization play important roles, as regulators of this process are proteins of the Rho family (e.g., RhoA, Rac1), which are also required for many cellular functions beyond cell mobility and cytokinesis [41]. These proteins, e.g., RhoA and Rock, play an important role in the MK endomitotic process [42,43]. Rock inhibitor treatments in mature MKs also lead to the downregulation of MYC, NFE2, MAFG, and MAFK transcript levels, suggesting that lowering the levels of these transcription factors is a prerequisite to driving the late stages of megakaryocyte maturation [44]. These transcription factors could be less important in the embryonic setting, therefore shifting the mode of platelet generation towards bud formation. Because Nfe2-deficient mice can form, release, and fragment proplatelets in vivo but are not able to form and release membrane buds [16], this could suggest that Nfe2 could be relevant for embryonic membrane budding.

In this study, we revealed the physiological characteristics of thrombopoiesis in mouse embryos, which is very different from adult thrombopoiesis. Using flow cytometry, we quantified platelet and reticulated platelet numbers at different developmental ages. We found that the overall platelet count is lower in embryonic compared to adult mice, yet the fraction of reticulated platelets was particularly high in the embryo. Platelet counts were particularly low at E11.5, but expanded with development. As megakaryocytes and platelets are probably more mature towards later developmental ages, we also found a decrease in the reticulated platelet fraction over time. In a fetal thrombosis model, platelets presented hyporeactivity during early fetal development. Specifically, they showed less adhesion and spreading capacity as well as reduced *p*-selectin mobilization compared with adults [45].

We quantified the MK numbers and overall FL cell numbers and found that MKs are a small fraction among FL cells. However, absolute cell numbers of MKs also increase within the FL niche over time. As the embryo develops, the ploidy of MKs increases, and 8N/16N stages appear. The FL undergoes a complex developmental process, originating from the ventral foregut endoderm [46] and differentiating into liver and biliary cells, fibroblasts, and stellate cells, with phases of accelerated growth as it is vascularized and colonized by hematopoietic cells to become the major fetal hematopoietic organ [34]. According to previous reports, immature or pre-hematopoietic stem cells (HSCs) from the YS colonize the FL, where definitive hematopoiesis takes place through HSC expansion (up to 40-fold [47]) and subsequent lineage commitment [48]. Beyond the FL, the spleen is also a site of hematopoiesis colonized by HSCs [49,50]. Among the earliest lineage commitments in the YS and FL are macrophages and MK progenitors [35,51]. These MK progenitors can apparently give rise to MKs in vitro, with a greater magnitude of in vitro expansion of FL progenitors than from adult marrow progenitors [52]. These MK progenitors then give rise to MKs. Although MKs were only a small fraction of total FL cells, their numbers increased substantially over time and gained in ploidy, which could represent the necessary MK maturation to achieve the aforementioned platelet expansion.

MKs can be generated from both primitive or definitive hematopoiesis [45]. In c-Myb knockout embryos, megakaryopoiesis and thrombopoiesis were not altered, and MKs remained abundant in the FL, indicating their independence from definitive hematopoiesis. This supports the concept that MKs derive from erythro-myeloid progenitors at this stage of development [22]. While we did not address potential compensatory mechanisms of primitive thrombopoiesis that might exist in the absence of definitive hematopoiesis, similar reports showed that c-Myb-deficient embryos had normal platelet numbers at E12.5 but became thrombocytopenic by E15.5 [44]. In summary, our findings in c-Myb-deficient embryos indicate that early platelet production is largely independent of definitive hematopoiesis. However, this transcription factor is required for sustaining definitive thrombopoiesis in the mature fetus.

## 5. Conclusions

We established a new protocol for intravital, real-time three-dimensional imaging, MP-IVM, to visualize embryonic thrombopoiesis in hematopoietic organs (YS, FL) across different stages of embryonic development. This technique, combined with flow cytometry, gave us the possibility to classify that at least two patterns of platelet shedding from MKs exist, either by an important process termed membrane budding or by conventional proplatelet release. The former seems to be the predominant form, which might be independent of c-Myb. The cellular and molecular mechanisms for bud release require further research.

## Figures and Tables

**Figure 1 cells-12-02411-f001:**
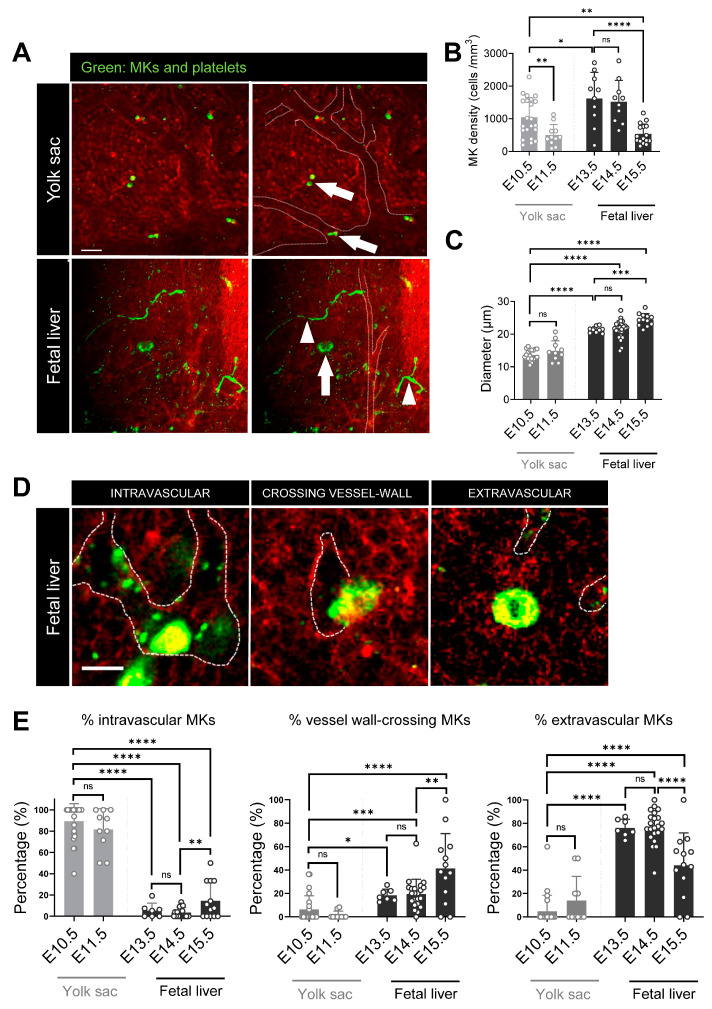
MK characterization in yolk sac and fetal liver. (**A**) Representative MP-IVM images of yolk sac (E10.5, scan depth: 25.5 µm) and fetal liver (E14.5, scan depth: 36 µm); white arrow, MKs without large protrusions; white arrow head, protrusion from MKs. Dashed line represents main vessel border. Bar, 50 µm. (**B**) MK density, E10.5 *n* = 22 (embryos), *n* = 8 (mothers); E 11.5 *n* = 11 (embryos), *n* = 7 (mothers); E 13.5 *n* = 10 (embryos), *n* = 6 (mothers); E14.5 *n* = 10 (embryos), *n* = 5 (mothers); E 15.5 *n* = 16 (embryos), *n* = 7 (mothers); and (**C**) MK diameter in yolk sac and fetal liver, E10.5 *n* = 20 (embryos), *n* = 7 (mothers); E11.5 *n* = 10 (embryos), *n* = 6 (mothers); E13.5 *n* = 9 (embryos), *n* = 6 (mothers); E14.5 *n* = 27 (embryos), *n* = 12 (mothers); E15.5 *n* = 12 (embryos), *n* = 5 (mothers). (**D**) Representative intravital images to show MKs position relative to the vasculature. MKs can be located intravascular (left, E15.5 fetal liver), crossing the vessel wall (middle, E15.5 fetal liver), or extravascular (right, E13.5 fetal liver). Bar, 20 µm. (**E**) Fraction of intravascular, vessel wall-crossing, and extravascular MKs in yolk sac and fetal liver. E10.5 *n* = 22 (embryos), *n* = 8 (mothers); E11.5 *n* = 11 (embryos), *n* = 7 (mothers); E13.5 *n* = 7 (embryos), *n* = 4 (mothers); E 14.5 *n* = 23 (embryos), *n* = 8 (mothers); E 15.5 *n* = 13 (embryos), *n* = 4 (mothers). **** *p* < 0.0001, *** *p* < 0.001, ** *p* < 0.01, * *p* < 0.05, or not significant (ns). Students’ *t*-test.

**Figure 2 cells-12-02411-f002:**
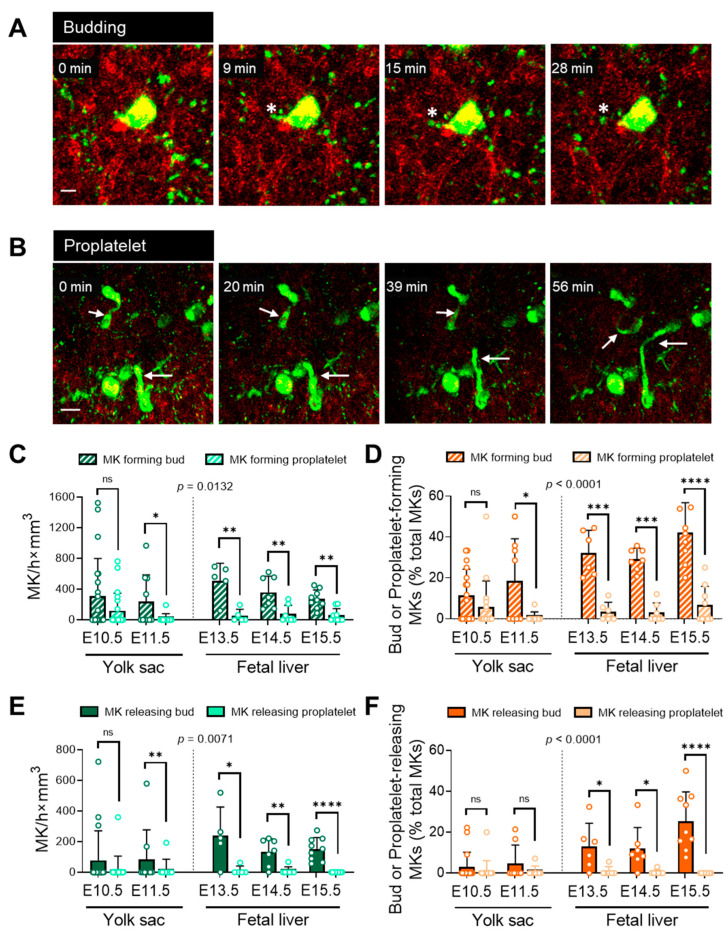
MK budding and proplatelet formation in the fetal liver and yolk sac. MP-IVM imaging of the fetal liver shows membrane budding ((**A**), bar 10 µm) and proplatelet formation ((**B**), bar 50 µm) of embryonic MK (E15.5 fetal liver). Percentage of bud and proplatelet-forming MKs (**C**) and releasing MKs (**D**) in fetal liver. (**E**) MK numbers releasing buds or proplatelets in the yolk sac and fetal liver. (**F**) Fraction of bud-releasing and proplatelet-releasing MKs among total MK population. **** *p* < 0.0001, *** *p* < 0.001, ** *p* < 0.01, * *p* < 0.05, or not significant (ns). Students’ *t*-test. E10.5 *n* = 21 (embryos), *n* = 7 (mothers); E11.5 *n* = 10 (embryos), *n* = 6 (mothers); E13.5 *n* = 6 (embryos), *n* = 5 (mothers); E14.5 *n* = 7 (embryos), *n* = 4 (mothers); E15.5 *n* = 9 (embryos), *n* = 5 (mothers).

**Figure 3 cells-12-02411-f003:**
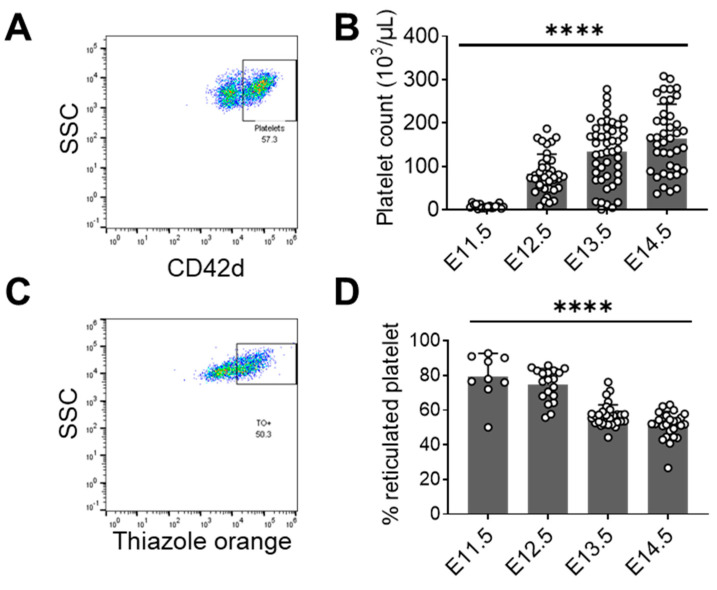
Flow cytometry analysis of embryonic platelets. (**A**) Representative CD42d (platelet marker) gating for platelet quantification after application of forward and sideward scatter gates, excluding doublets. Platelet population is indicated inside the square (**B**) Platelet count in embryonic blood from different embryos for each embryonic day; E11.5 *n* = 18 (embryos), *n* = 3 (mothers); E12.5 *n* = 38 (embryos), *n* = 4 (mothers); E13.5 *n* = 47 (embryos), *n* = 6 (mothers); E14.5 *n* = 40 (embryos), *n* = 6 (mothers); (**C**) Representative reticulated platelet gating strategy using thiazole orange (TO), with previous gating as described in 3A. Reticulated platelet population is indicated inside the rectangle (TO+); (**D**) reticulated platelet percentage in embryonic blood from different embryos for each embryonic day; E11.5 *n* = 9 (embryos), *n* = 3 (mothers); E12.5 *n* = 19 (embryos), *n* = 4 (mothers); E13.5 *n* = 32 (embryos), *n* = 6 (mothers); E14.5 *n* = 26 (embryos), *n* = 6 (mothers). **** *p* < 0.0001, one-way ANOVA test. (**B**,**D**) Open circles indicate number of embryos analyzed.

**Figure 4 cells-12-02411-f004:**
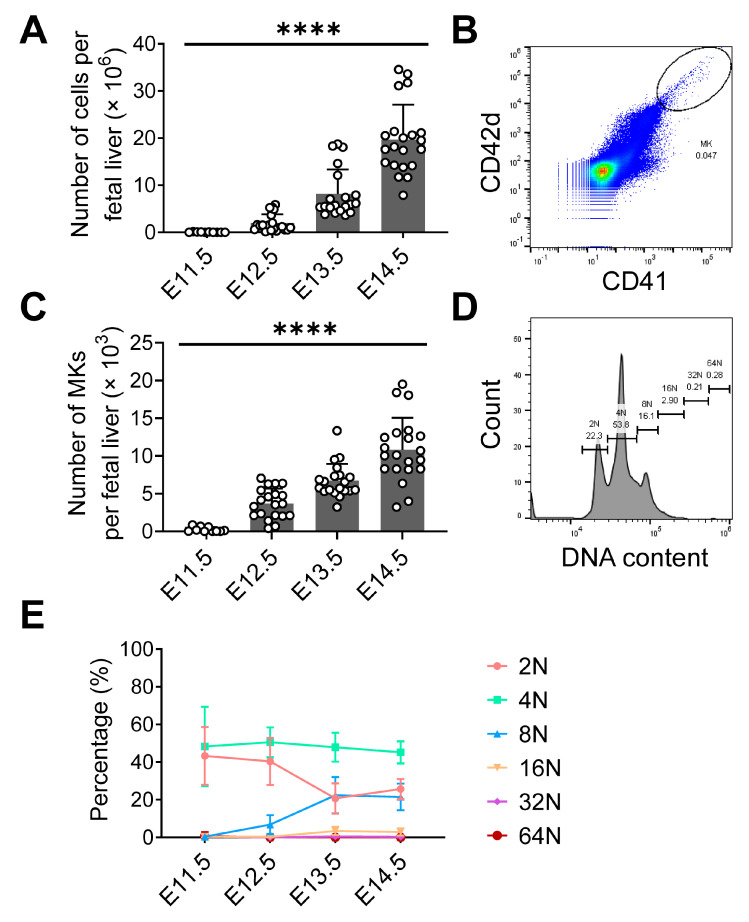
Flow cytometry analysis of fetal liver MKs. (**A**) Fetal liver cell numbers during embryonic development. (**B**) Representative MK gating strategy, MK population is indicated inside the circle. (**C**) MK numbers in fetal liver during embryonic development. (**D**) Representative MK ploidy analysis. (**E**) MK ploidy during embryonic development. **** *p* < 0.0001, one-way ANOVA test. E11.5 *n* = 10 (embryos), *n* = 3 (mothers); E12.5 *n* = 20 (embryos), *n* = 4 (mothers); E13.5 *n* = 20 (embryos), *n* = 4 (mothers); E14.5 *n* = 21 (embryos), *n* = 5 (mothers). (**A**,**C**) Open circles indicate number of embryos analyzed.

**Figure 5 cells-12-02411-f005:**
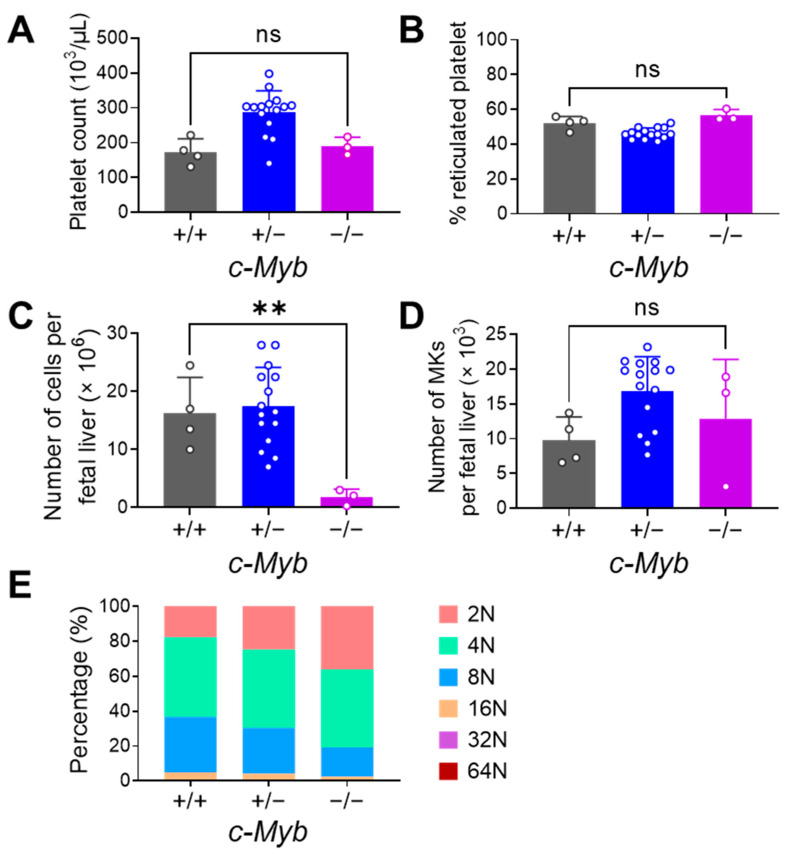
Flow cytometry analysis of platelets and MKs in c-Myb transgenic embryos at E14.5. (**A**) Platelet count in embryonic blood. There was a significant difference between c-Myb^−/−^ (190.0 ± 25.6 × 10^3^/µL) and c-Myb^+/−^ (287.3 ± 16.1 × 10^3^/μL) embryos. (**B**) Reticulated platelet percentage in embryonic blood. (**C**) Total cell numbers of fetal liver. (**D**) MK numbers in fetal liver. (**E**) MK ploidy analysis in fetal liver. ** *p* < 0.01, otherwise not significant (ns). Students’ *t*-test. E14.5 *n* = 4 (c-Myb^+/+^), *n* = 15 (c-Myb^+/−^), *n* = 3 (c-Myb^−/−^), *n* = 4 (mothers).

## Data Availability

Further information and requests for resources and data should be directed to and will be fulfilled by the lead contacts Christian Schulz and Mathias Orban.

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
