# Peer review of "Multiphoton In Vivo Microscopy of Embryonic Thrombopoiesis Reveals the Generation of Platelets through Budding"

_cells, 2023, doi:10.3390/cells12192411_

Round 1

Reviewer 1 Report

In this manuscript, Huan Liu et al established a new protocol for intravital, real-time three-dimensional imaging of embryonic thrombopoiesis in haematopoietic organs (yolk sac, fetal liver) by using multiphoton intravital microscopy (MP-IVM). They found that fetal liver MKs provide higher thrombopoietic activity than yolk sac MKs, and the fetal platelets generated from MKs either by membrane budding or proplatelet release. They found the membrane budding seems to be the predominant form of platelets release which might be independent of c-Myb. The new technique of MP-IVM for direct visualization of embryonic thrombopoiesis in vivo is impressive. However, I have the following concerns.

Major:

1.         In Line 80, Page 2, the author used “c-Myb transgenic mice to determine the association of platelet production with early hematopoiesis”, however, in the result part the author used “c-Myb deficient embryos”, did they refer to the same strain of c-Myb knockout mice? In regarding to the platelet counts and the MK numbers, is there any significant differences between c-Myb+/- and WT or c-Myb-/- mice? Did the authors used littermate mice?

2.         Figure legends for fig 2E and fig 2F were missing. What is the difference between MK forming bud vs releasing bud? Why the significant difference was not consistent in YS in fig 2D and 2F?

3.         In fig 3, it was confusing how the authors proved that it was the bud-release from MKs that translated into an increase of peripheral platelets (Line 345, page 9).

4.         The authors should provide the gating strategy for flow cytometry in Figure 3A and 3C, similar for Figure 4B.

5.         In Figure 5E, there is a higher proportion of high-ploidy MKs (≥4N) in the c-Myb+/+ group, suggesting a potential vitality for platelets generation. However, in Figure 5A, the peripheral blood platelet counts in the c-Myb+/+ group are comparable to the c-Myb-/- group and lower than the c-Myb+/- group. Please explain.

6.         From the results presented in Figure 5, c-Myb knockout may affect the total cell count in the fetal liver, indicating it’s effect on fetal liver development. Therefore, is was possible that c-Myb may impact embryonic thrombopoiesis via fetal liver niche.

Minor:

1.      The abbreviations "MEPs" was mentioned repeatedly in line 41 and line 51.

2.      Genotypes of mice throughout the manuscript should be superscripted, such as Myb-/-.

3.      All the figures (from A to E) should be referred in Results 1 section.

4.      In line 348, the platelet count should be represented as (8.1 ± 4.6) × 103/µL, keep consistent throughout the entire manuscript.

The quality of English language is good.

Reviewer 2 Report

In the study "Multiphoton in vivo microscopy of embryonic thrombopoiesis reveals the generation of platelets through budding”, Liu et al. investigate embryonic thrombopoiesis by in multiphoton in vivo microscopy with a specific focus on proplatelet formation and budding. The authors analyze MK density, distribution and platelet formation in the yolk sac and fetal liver at different embryonic days. In addition, mice lacking c-Myb, a transcription factor involved in definite hematopoiesis. The study rather descriptive.

I have a few comments which need to be addressed before I can recommend publication of this manuscript.

-        Please include the process of platelet budding with references in your introduction

-        Figure 1:

o  1B: How long were the videos acquired during MP-IVM? How long was each single MK observed? How did you calculate the fraction?

o  In your later analysis, did you image every position for 1 h(MK/h/mm³)?

o  What is the size of the analyzed area? The variability of the data is quite huge. Did you extrapolate to 1 mm³? This might increase the counting error.

o  How do you explain the huge SD? How many animals were analyzed (please indicate in every figure legend! -> display as mean per animal and indicate the n-number in each figure legend)

o  How do you explain the high density of the MKs?

o  C: how were these data analyzed? how many animals were analyzed? you need to calculate the mean per mks per animal (single points) and then the mean per day

-        Figure 2:

o  the numbers per h are much lower compared to figure 1 MKs/mm³ -> how do you explain this? how long did you observe in fig 2? For 1 h ore did you extrapolate? Maybe extrapolation is overestimating the amount per mm³? how much do these mk move within your observation period?

o  As mentioned above, please indicate: how is the data displayed (mean/median/SD/SEM)? since the variation is quite high.

o  please discuss: in YS, most MK do not form any platelets, neither via budding nor via PPF

o  When does PPF during embryonic development start?

o  Adjust figure legend: E, F missing

-        Figure 3: how do embryonic platelets look like in the FSC/SSC scatter blot? Please include it. Thiazole orange staining: On basis of what was the gate drawn? This looks a bit arbitrary.

-        L. 374 ff: If these were different embryos for different analysis days, what I would guess, please rephrase. This sentence sounds like one animal was analyzed over time.

-        what is a marker for primitive hematopoiesis? Your data suggest a role of primitive hematopoiesis for platelet count and the formation of reticulated platelets. How would a respective ko  mouse model behave in your assays?

-        Please check the manuscript for wording/sentence structure (e.g. l. 93, l. 425 ff, etc.)

Round 2

Reviewer 1 Report

The reviewer has no more comments and suggestions for the authors.

Author Response

We thank the reviewer for his positive evaluation

Reviewer 2 Report

I thank the authors for extensively addressing my questions.

- There's a typo in the heading 2.2: intravital

- besides otherwise stated,  my previous comment to l. 374 ff. (now l. 399) has not been addressed yet. Please adapt the wording accordingly.

Author Response

We thank the reviewer for this positive comments.

We have corrected the typo in 2.2. heading.

For the remaining  comment to l. 374 ff. (now l. 399), we have changed the wording accordingly:

Between E11.5 and E15.5, the fetal liver undergoes a period of accelerated growth with rising FL cell numbers and density [34]. FLs from embryos of different ages (n=10-21 embryos and n=3-5 mothers) were analyzed, showing an increased total cell number from 0.04 ± 0.05 × 106 cells at E11.5 to 19.7 ± 7.4 × 106 cells at E14.5 per FL (Figure 4A). 

Using flow cytometry, we quantified MKs in FL (Figure 4B) in different embryonic stages. Absolute MK numbers per FL were calculated from the FL cell count multiplied with the fraction of MK in FL. During FL development, total cell numbers strongly increased. In parallel, also fetal MK numbers increased significantly (Figure 4C). Absolute MK counts grew from 0.2 ± 0.3 × 103 at E11.5 to 10.8 ± 4.2 × 103 at E14.5 (n=10-21 embryos and n=3-5 mothers).